# Broad-Spectrum In Vitro Activity of *N*α-Aroyl-*N*-Aryl-Phenylalanine Amides against Non-Tuberculous Mycobacteria and Comparative Analysis of RNA Polymerases

**DOI:** 10.3390/antibiotics13050404

**Published:** 2024-04-28

**Authors:** Markus Lang, Uday S. Ganapathy, Rana Abdelaziz, Thomas Dick, Adrian Richter

**Affiliations:** 1Institut für Pharmazie, Martin-Luther-Universität Halle-Wittenberg, Kurt-Mothes-Straße. 3, 06120 Halle (Saale), Germany; markus.lang@pharmazie.uni-halle.de (M.L.); rana.abdelaziz@pharmazie.uni-halle.de (R.A.); 2Center for Discovery and Innovation, Hackensack Meridian Health, 111 Ideation Way, Nutley, NJ 07110, USA; uday.ganapathy@hmh-cdi.org; 3Department of Medical Sciences, Hackensack Meridian School of Medicine, 123 Metro Boulevard, Nutley, NJ 07110, USA; 4Department of Microbiology and Immunology, Georgetown University, 3900 Reservoir Road, Washington, DC 20007, USA

**Keywords:** *N*α-aroyl-*N*-aryl-phenylalanine amides, RNA polymerase, *M. abscessus*, NTM, non-tuberculous mycobacteria, MMV688845, AAPs

## Abstract

This study investigates the in vitro activity of *N*α-aroyl-*N*-aryl-phenylalanine amides (AAPs), previously identified as antimycobacterial RNA polymerase (RNAP) inhibitors, against a panel of 25 non-tuberculous mycobacteria (NTM). The compounds, including the hit compound MMV688845, were selected based on their structural diversity and previously described activity against mycobacteria. Bacterial strains, including the *M. abscessus* complex, *M. avium* complex, and other clinically relevant NTM, were cultured and subjected to growth inhibition assays. The results demonstrate significant activity against the most common NTM pathogens from the *M. abscessus* and *M. avium* complexes. Variations in activity were observed against other NTM species, with for instance *M. ulcerans* displaying high susceptibility and *M. xenopi* and *M. simiae* resistance to AAPs. Comparative analysis of RNAP β and β′ subunits across mycobacterial species revealed strain-specific polymorphisms, providing insights into differential compound susceptibility. While conservation of target structures was observed, differences in compound activity suggested influences beyond drug–target interactions. This study highlights the potential of AAPs as effective antimycobacterial agents and emphasizes the complex interplay between compound structure, bacterial genetics, and in vitro activity.

## 1. Introduction

Within the field of infectious diseases, the exploration of mycobacterial infections has always revolved around the formidable presence of *Mycobacterium tuberculosis* (Mtb), rightfully claiming its place as a global health concern [1]. However, amidst this predominant focus, a group of less discussed yet clinically relevant entities is emerging—non-tuberculous mycobacteria (NTM) [2,3,4,5]. NTM, comprised of over 190 species, are ubiquitous in the environment, inhabiting soil, water sources, and various organic substrates. These mycobacterial species, distinct from their tuberculosis-causing congener, have garnered increasing attention in recent years due to their diverse clinical manifestations and growing significance in healthcare settings worldwide [6].

While historically deemed harmless environmental dwellers, their potential to cause diseases in immunocompromised populations and individuals with chronic respiratory conditions like cystic fibrosis [7,8,9,10] or bronchiectasis [11] has highlighted their clinical relevance. NTM infections manifest across a spectrum of clinical presentations that includes pulmonary, cutaneous [12,13], and disseminated forms. Often characterized by a protracted and subtle onset, these infections cause diagnostic dilemmas that lead to delays in appropriate therapeutic interventions, thereby amplifying patient morbidity and mortality rates. 

Epidemiological studies worldwide unveil a rising trend in NTM-associated diseases [14,15,16], particularly among immunocompromised populations and individuals with chronic respiratory conditions. Moreover, advancements in diagnostic modalities, including molecular techniques and improved culturing methodologies, have unveiled a previously underestimated burden of NTM infections, underscoring the need for heightened clinical vigilance and a deeper understanding of their pathogenic mechanisms.

While the majority of NTM-related pulmonary infections are attributed to species such as the *Mycobacterium avium* complex (MAC) [10,17,18], and *Mycobacterium abscessus* complex (MABC) [19,20], there exist lesser-known species that sporadically provoke pulmonary manifestations. Their occurrence and distribution exhibit notable regional disparities, reflecting diverse environmental reservoirs and varying host susceptibilities across geographical regions [16,21]. Understanding the clinical relevance of these less common NTM species in pulmonary infections is crucial, especially in instances where conventional diagnostic tests may fail to identify the causative organism promptly. Their infrequent occurrence underscores the importance of vigilance among clinicians and microbiologists to consider these NTM species in the differential diagnosis of chronic or refractory pulmonary conditions.

Beyond their clinical impact, the management of NTM infections presents a formidable challenge. Their innate resistance to most anti-tubercular agents and other antibiotics, coupled with variations in susceptibility profiles among different species, necessitates tailored therapeutic regimens based on accurate identification and susceptibility testing as well as new antimycobacterial drugs to secure the treatment of patients.

*N*α-aroyl-*N*-aryl-phenylalanine amides (AAPs) represent a compound class that has displayed promising activity against *Mtb* and *Mycobacterium abscessus* [22,23,24], offering a potential avenue for novel therapeutic interventions against these challenging infections. Research exploring the medicinal chemistry [25,26,27] and antimycobacterial properties [28] of AAPs has shown encouraging results in vitro, demonstrating their ability to inhibit the growth of different mycobacterial species. This compound class’s mechanism of action targets the essential mycobacterial RNA polymerase (RNAP) [29], disrupting key cellular processes and counteractions vital for their survival and propagation.

This study attempts to delve into the field of clinically relevant NTM species and their susceptibility to the promising compound class of AAPs. Therefore, a selection of active AAPs was tested against a broad panel of NTM to evaluate the therapeutic potential of the substance class across a broader range of NTM infections. We focused largely on type strains that are available from culture collections to allow for comparability. In addition to the published activity data of AAPs against *M. abscessus* subsp. *abscessus*, the two other subspecies of *M. abscessus,* subsp. *massiliense* and subsp. *bolletii*, are evaluated as well as a panel of clinical isolates of the *M. abscessus* complex. For the *M. avium* complex, two different strains of *M. avium* subsp. *hominissuis* (the most virulent *M. avium* subspecies for humans), *M. intracellulare*, and *M. chimaera* were tested. The set selected for mycobacteria that have occasionally emerged as human pathogens consists of their respective laboratory-type strains. It also contains the two soft tissue pathogens *M. marinum* and *M. ulcerans*, the causative agents of fish tank granuloma [30] and Buruli ulcer [31], respectively. A comparative approach serves as a foundation for unraveling the relationship between bacterial genomic diversity and antibiotic responses, ultimately contributing to the advancement of targeted antimicrobial therapies in the face of evolving mycobacterial resistance.

## 2. Results and Discussion

### 2.1. Selection of Nα-Aroyl-N-Aryl-Phenylalanine Amide Compounds

The compounds selected for assessment against the strains described in this study were synthesized and characterized as described previously [25,32]. The selection of the test set considered their activity against previously tested mycobacteria, as well as diverse structural features that could affect the activity against different NTM species. Compounds **1** to **7** were synthesized during a detailed SAR study that varied the ring systems of the chemical scaffold in the search for better activity against NTM. The morpholine moiety was substituted by thiomorpholine sulfoxides and sulfones, which proved advantageous for the activity and solubility of the compound class, in addition to the exchange of the thiophene carboxylic acid amide to 2-fluorobenzoic acid amides. Compound **6** was included because it showed high activity while harbouring the hydroxyl group [25]. We also wanted to determine the influence of 5-fluoro substitution in para-position to the morpholine moiety (compounds **4** to **6**). Compounds **8** to **10** showed high anti-NTM activity and have been synthesized and tested in a previous study with the aim to improve the stability of the compound class by additional sterical hindrance of the amide bonds [32]. The molecular structures of the tested compounds are given in Figure 1 while their previously published activities against NTM can be found in Table 1. For comparative purposes, the initial hit compound of phenylalanine amides, MMV688845 (MMV), was included. Additionally, clarithromycin (CLR) was used as a positive control due to its clinical relevance.

### 2.2. Inhibition of M. abscessus Complex

All the AAPs that were tested displayed activity against the selected *Mycobacterium abscessus* complex strains. An overview of the calculated MIC_50_ values is given in Table 2. As microbial populations are often diverse and different strains may respond differently to antimicrobial agents under assay conditions, we decided to utilize the MIC_50_ as a comparative measure of activity, because in some cases, MIC_50_ values may reflect a more representative and comparable average than MIC_90_ values, as the latter can be influenced by outliers and/or growth and plate effects [33]. The respective MIC_90_ values for each compound and strain calculated from the same data sets are displayed in the Appendix A.

Testing CLR and the AAP hit compound MMV resulted in MIC_50_ values well comparable to those reported in the literature [22,33] (see Figure 1 for exemplary dose–response curves). The obtained MIC_50_ values of the AAPs are generally in the low micromolar concentration range between 0.1 µM and 2.1 µM. While the MIC_50_ values against subsp. *abscessus* were comparable to those against subsp. *massiliense*, there was a pattern of slightly lower activities against subsp. *bolletii,* which was also reported for other antibiotics [34]. Structures containing a sulfone moiety generally exhibit lower MIC_50_ values, with compound **8** displaying the highest activity against all subspecies (subsp. *abscessus*: 0.2 µM, subsp. *massiliense*: 0.1 µM, subsp. *bolletii*: 0.3 µM) translating to a 7-19-fold enhancement in activity against the various subspecies when compared to the hit compound MMV. Compound **7** and **8** that exhibited promising in vitro activity (MIC_90_ of 0.78 µM) against *M. abscessus* subsp. *abscessus* ATCC 19977 were further investigated against a range of clinical isolates of the *M. abscessus* complex, expanding the evaluation to a broader range of genotypes within this species (Table 3). The results demonstrated a comparable potency to the type strains, confirming their potential as effective antimicrobial agents for the treatment of *M. abscessus* infections.

### 2.3. Inhibition of M. avium Complex

The examined AAPs demonstrate substantial activity against the selected *M. avium* complex strains, with the determined MIC_50_ values being in line with those against the *M. abscessus* complex (Table 4). A promising observation is that these derivatives show potent activity against the *M. avium* complex, with a 5-10-fold improvement in in vitro activity over the hit compound MMV688845. The high clinical relevance of *M. avium* complex infections, for which rifampicin’s therapeutic benefits are controversial, emphasizes the potential of AAPs as novel RNAP inhibitors [29,35].

### 2.4. Inhibition of Other NTM

The panel of AAPs was tested against a variety of NTM that occasionally occur as human pathogens. The respective MIC_50_ values are displayed in Table 5. The selection showed activities in a comparable range to those seen against the *M. abscessus* complex and *M. avium* complex. Notable differences were seen for *M. xenopi* and *M. simiae*, against which activities were reduced. While clarithromycin showed an MIC_50_ of 0.03 µM against *M. xenopi*, *M. simiae* was less susceptible (MIC_50_ 10.7 µM) to clarithromycin, which is consistent with what has been reported in the literature [36,37,38]. *M. ulcerans* showed a high susceptibility to AAPs with MIC_50_ values as low as 10 nM. A further difference is that certain mycobacterial species react differently to **6**, the only compound in which phenylalanine is replaced by tyrosine. The presence of an additional hydroxyl group reduces the activity in *M. kansasii*, *M. malmoense*, *M. marinum* and *M. szulgai*, with activities always lower than those of the hit compound MMV. For *M. ulcerans* and *M. xenopi*, **6** shows the highest activities among all tested compounds. This suggests that a higher polarity of the compounds may be advantageous for in vitro activity against these strains.

### 2.5. Comparative Analysis of RNAP β and β’ Subunits

In this study, we employed comparative alignment analysis to investigate the genomic diversity among the selected mycobacterial strains and elucidate whether alterations of the protein primary structure within the binding pocket of AAPs could explain the observed differences in AAP susceptibility. By integrating the primary target sequences of the β and β′ RNAP subunits from the mycobacterial strains in protein–protein alignment (performed within positions 450–600 and 800–880 for the β and β′ subunits, respectively, the complete alignment in this area can be found in the Appendix A), we identified strain-specific amino acid variations in comparison to the reference sequence of a published protein structure of *Mtb* RNAP that was co-crystallized with the AAP [29] analog D-AAP1 (PDB: 5UHE). An overview of the contacts of D-AAP1 and its RNAP binding site based on PDB: 5UHE is given in Figure 2. AAPs are highly active against the *Mtb*-type strain ATCC 25618 H37Rv [22,24,25]. The results of the alignment analysis are displayed in Figure 3. The table was constrained to show amino acid variations located within a 7 Å distance of target-bound D-AAP1 [29] to the surrounding amino acids to limit the analysis to the area around the binding site. *Mtb* RNAP positions that did not show variations for any strain were excluded from the depiction as well as strains that did not show any polymorphisms in these areas.

The overall sequence identity of the *Mtb* β subunit to the NTM β subunits is high (89% to 95%), with the trend that the fast-growing mycobacteria (*M. chelonae*, *M. fortuitum* and *M. abscessus*) show lower identity values (89–91%), while the skin pathogens *M. ulcerans* and *M. marinum* have 95% identity each and even 99% identity in the region that contains the AAP binding site (positions 450–600). *M. chelonae*, *M. fortuitum*, *M. xenopi* and *M. abscessus* all display an alanine-to-glycine variation at position 565 of the β subunit in close proximity to the AAP binding site. However, this variation does not appear to affect their susceptibilities possibly due to the minor differences in volume and polarity between alanine and glycine. In *M. fortuitum*, a leucine to methionine exchange (Grantham’s distance 15, a measure of the similarity of amino acids in protein structures that combines the composition, volume and polarity for comparison, in which small values indicate high similarity and high values indicate low similarity [39]) occurs at position 560, which does not seem to result in lower activities in this case. This particular position reportedly resulted in a resistant *M. abscessus* Bamboo strain after a leucine to proline exchange [28] (Grantham´s distance 98). The leucine at position 560 ensures a transition to a random coil formation (R562-V568) that is in direct contact with AAP structures and is therefore crucial for the right orientation of the binding pocket. As proline disrupts secondary structures, this variation could cause a conformational change in the binding site that cannot be compensated. Additionally, *M. fortuitum* and *M. xenopi* display a proline to serine exchange at position 477, which is an essential lipophilic binding contact to AAP’s anilide aromatic system and the phenylalanine aromatic system. Its importance for the interaction was demonstrated by the formation of a resistant mutant after a proline to leucine variation [28]. Leucine exhibits similar lipophilic properties as the proline side chain, but its higher spatial demand causes clashes with the anilide aromatic system and the random coil formation (R562-V568), altering the arrangement of the binding pocket and leading to AAP resistance. The proline–serine exchange of *M. fortuitum* and *M. xenopi* leads to a binding pocket that is less hydrophobic but has a similar volume to the native proline conformation, resulting in AAP activity. However, the binding of AAPs could be restricted due to the differences in polarity, which might be a part of the explanation for the slightly lower activities in *M. fortuitum* and the loss of activity against *M. xenopi*.

Comparing the sequence identities of the *Mtb* β′ subunit with their NTM counterparts, we observed high degrees of homology (90–97% sequence identity), whereas the lowest value of 90% was found for *M. abscessus* and the highest values were again found for *M. ulcerans* and *M. marinum*, which showed 97% each. The β’ subunit of *Mtb* shows two clusters of amino acids that build up the binding surface to AAPs. Only two variations were observed within a 7 Å distance to the AAP binding site. A prominent variation that all displayed NTM strains exhibit is the valine to isoleucine exchange at position 836. This position is close to the aryl carboxylic acid amide structure of the AAPs and contributes to the lipophilic surface that interacts with the aromatic system. We do not expect that this amino acid exchange affects activities, as the additional methylene group in isoleucine is not largely affecting the properties of the binding pocket in the matter of polarity. The additional expansion does not appear to influence binding and activity. The only other variation present was found in *M. abscessus* ATCC 19977, where phenylalanine 831 is changed to tyrosine (Grantham’s distance 22). However, this alteration does not seem to affect the activity against *M. abscessus* ATCC 19977. The side chain of the phenylalanine is oriented away from the AAP binding site into an unrelated, open cleft resulting in no direct interaction. Phenylalanine and tyrosine share similar properties, making it probable that the same is true for tyrosine. Still, the exchange could lead to differences in the geometry of the binding pocket due to its proximity to the binding site.

The aromatic system of the phenylalanine part of AAPs extends into a lipophilic cleft of the β subunit. One of the constituents of the surface of this cleft is proline at position 477, which was previously discussed for *M. fortuitum* and *M. xenopi*. This amino acid is in close proximity to the *para* position of the phenylalanine group of AAPs. In *M. xenopi*, the exchange from proline to serine could induce a geometric shift that provides an additional hydrogen bond between the serine backbone and the tyrosine hydroxyl group. This could explain the higher potency of **6**. However, for the strains that show reduced susceptibility to **6**, no variations in the lipophilic cleft were found, making it challenging to explain the difference in activity.

## 3. Materials and Methods

### 3.1. Bacterial Cultures and Strains

For general bacteria culturing and inhibition experiments, Middlebrook 7H9 broth (BD Difco) was supplemented with 0.5% albumin, 0.2% glucose, 0.085% sodium chloride, 0.0003% catalase, 0.2% glycerol, and 0.05% Tween 80^®^. Most bacterial strains analyzed in this study were purchased as type strains either from the American Type Culture Collection (ATCC) or the Culture Collection University of Goteborg (CCUG) as indicated. *M. abscessus* subsp. *abscessus* Bamboo was isolated from the sputum of a patient with amyotrophic lateral sclerosis and bronchiectasis and was provided by Wei Chang Huang, Taichung Veterans General Hospital, Taichung, Taiwan. Clinical isolates covering the *M. abscessus* complex (M9, M199, M337, M404, M422, M232, M506, and M111) were provided by Jeanette W. P. Teo (Department of Laboratory Medicine, National University Hospital, Singapore). Detailed information on the origin of these isolates is given in reference [40]. *M. avium* subsp. *hominissuis* strain 109 (MAC109) was isolated from the blood of a patient with AIDS and was provided by Petros C. Karakousis (Johns Hopkins University) [41]. *M. avium* subsp. *hominissuis* strain 11 originates from the bone marrow of an AIDS patient suffering from a disseminated infection caused by *M. avium* [22,42]. The isolate was provided by Jung-Yien Chien and Po-Ren Hsueh, National Taiwan University Hospital, Taipei.

### 3.2. Growth Inhibition Assay

Growth inhibition assays were performed in 96-well plate format. The wells were filled with 100 µL of supplemented 7H9 medium before dispending 10 mM compound stock solutions in DMSO into the wells using a Tecan D300e digital dispenser. For each compound, a 10-point 3-fold dilution series or a 10-point 2-fold dilution series was prepared that typically started at a concentration of 100 µM. The DMSO concentrations were normalized to 2%. The cultures of the respective bacterial strains were grown to the mid-log phase, which was indicated by a measured OD_600_ between 0.4 and 0.6. A sufficient aliquot of the culture was taken from the culture and diluted to an OD_600_ of 0.1 with fresh 7H9 medium (1 × 10^7^ CFU/mL). Next, 100 µL of the resulting bacterial suspension was used to inoculate the prefilled wells, which resulted in a total volume per well of 200 µL with an OD_600_ of 0.05 (5 × 10^6^ CFU/mL, 1% DMSO). Each plate included 8 untreated wells containing 1% DMSO and 8 sterile wells for blank corrections. The plates were sealed with Parafilm^®^ (Bemis Company, Nennah, WI, USA), wrapped in damp paper towels, and placed in tight-closing plastic boxes, before incubation at 37 °C and shaking at 110 rpm. Fast-growing NTM (*M. abscessus* and subsp., *M. fortuitum* and *M. chelonae*) were incubated for 3 days as a standard procedure, while the slow-growing strains (all the other strains) were incubated for 5 days. Due to its particularly slow growth rate, *M. ulcerans* was incubated for 10 days.

### 3.3. Determination of MIC Values

To determine the minimal inhibitory concentration at 50% growth inhibition relative to an untreated control (MIC_50_), OD_600_ values of each well were measured with a Tecan Infinite M200 plate reader on day 0 and day 3, day 5 or day 10. Before measuring the OD on the final day of analysis, the sedimented bacterial cells were resuspended with either manual pipetting or with the use of an Eppendorf epMotion 5070 pipetting robot. On day 0 and the final day of analysis, the average OD of the sterile wells was subtracted from the remaining wells for blank correction. To generate the bacterial growth values for every well, the blank-corrected day 0 values were subtracted from the blank-corrected day 3/5/10 values. For each compound, the growth values of the two corresponding untreated wells gave the average drug-free growth, which is equal to 100% growth/0% inhibition. To calculate the % growth of each drug-containing well, their growth values were related to their respective drug-free growth values. GraphPad Prism 10.0 was used for graphical analysis, curve fitting and calculations. Dose–response curves were plotted with % inhibition (=100% growth) versus compound concentration. For the calculation of MIC values, the obtained data points were fitted utilizing a standard variable slope Hill function (bottom asymptote value constrained to equal 0). The resulting function was used to calculate MIC_50_. The values were calculated from two technical replicates and averaged for each compound.

### 3.4. Protein–Protein Primary Structure Alignment

To compare the primary structures of the RNAP β and β′ subunits of different NTM, we utilized the protein–protein BLAST algorithm provided by NIH (https://blast.ncbi.nlm.nih.gov/Blast.cgi (accessed on 4 April 2024). The primary structures of the type strains were obtained from the Pathosystems Resource Integration Center (PATRIC) database, which is provided by the Bacterial and Viral Bioinformatics Resource Center (BV-BRC, University of Chicago, https://www.bv-brc.org/ (accessed on 4 April 2024)). Additionally, we retrieved the genome of MAC109 from Matern et al. [41]. The comparative genomics analysis excluded *M. abscessus* subsp. *massiliense*, *M. abscessus* subsp. *bolletii*, *M. abscessus* clinical isolates and *M. avium* subsp. *hominissuis* strain 11.

### 3.5. Visualization of Protein Models

For surface and interaction analysis as well as visualization of the *Mtb* RNAP 3D structure (PDB: 5UHE [29]), we utilized UCSF ChimeraX (Resource for Biocomputing, Visualization, and Informatics at the University of California, San Francisco, CA, USA) [43], as well as the Maestro graphical interface (Schrödinger Release 2022-3: Maestro, Schrödinger, LLC, New York, NY, USA, 2021).

## 4. Conclusions

AAPs that emerged as RNAP inhibitors against *Mtb* show promising in vitro activity against a wide range of NTM. In addition to the *M. abscessus* complex and the *M. avium* complex, we demonstrated that AAPs show in vitro activity against a less common yet clinically relevant set of NTM. The majority of NTM were susceptible to AAPs and particularly high activities were observed against *M. marinum* and *M. ulcerans*, while *M. simiae* and *M. xenopi* showed a lower level of susceptibility. The various AAPs that were tested exhibit comparable inhibition tendencies across different mycobacterial species. The data obtained from these other mycobacteria align well with the previously published structure–activity relationships. The comparative analysis of the target sequences of different mycobacterial species, focusing on the binding pocket of AAPs, revealed a high degree of conservation in both the primary and spatial structure within the relevant areas of the β and β’ RNAP subunits, showing the potential value of AAPs as broad-spectrum anti-mycobacterial inhibitors. Variations in in vitro activities were observed among compounds with specific structural elements, such as the *para*-hydroxy group in tyrosine (**6**). However, the observed polymorphisms did not uniformly align with alterations in compound susceptibility, underscoring the multifaceted nature of drug–bacteria interactions.

The study emphasizes the potential of AAPs as versatile antimycobacterial agents. However, the variations in compound activity across different strains indicate the need for further exploration into the interplay between compound structure and bacterial physiology. The study’s conclusions are limited by the small number of strains used for each species. Follow-up work should include testing a larger panel of clinical isolates to verify the results, especially for the species that showed conspicuous features. This study offers valuable insights into the susceptibility of NTM to AAPs and provides a basis for the development of more effective treatments against a wide range of mycobacterial infections.

## Data Availability

The raw data supporting the conclusions of this article will be made available by the authors upon request.

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
