# Peer review of "Broad-Spectrum In Vitro Activity of Nα-Aroyl-N-Aryl-Phenylalanine Amides against Non-Tuberculous Mycobacteria and Comparative Analysis of RNA Polymerases"

_antibiotics, 2024, doi:10.3390/antibiotics13050404_

Round 1

Reviewer 1 Report

Comments and Suggestions for Authors

The manuscript entitled "Broad-Spectrum In Vitro Activity of Nα-aroyl-N-aryl-Phenylalanine Amides Against Non-Tuberculous Mycobacteria and Comparative Analysis of RNA Polymerases" reports in vitro activity of Nα-aroyl-N-aryl-phenylalanine amides (AAPs) against a panel of 25 non-tuberculous mycobacteria (NTM) as well as the in silico binding and sequence analyses. The study is well designed; however, some amendments are required in the presentation of results.

- Table 1: "M. tuberculosis H37Rv CCUG 48898-T" should be revised because CCUG 48898-T is not M. tuberculosis H37Rv.

- "subspecies" should be abbreviated as "subsp." and should not be written italic. Also, the strain names, codes etc. should not be written italic.

- Table 3: "M. abscessus" should not be abbreviated as "Mabs".

- Table 3-5: The color codes are confusing. Same values were colored in different shades.

- Table 6: Reference species should be defined.

Reviewer 2 Report

Comments and Suggestions for Authors

The article is of interest and I particularly enjoy the effort of the authors to correlate activity with the allelic changes of the target althoug, as stated by the authors, other strain-specific elements can also modify the activity of the antibiotic.

The article has a fundamental problem that, however is easy to solve. MIC cannot be presented as averages. By definittion MIC is the MInimal inhibitory concentration. This means that if, in two assays, the values differ, MIC is the lowest value, not the average. This has some exceptions. If there are doubts, MICs can be repeated and if, for instance 1 value is low and 4 are high, one can think that the low value is a mistake and the right one is the high, buy never the average.

The authors would like to modify the MIC values all along the article taking this observation into consideration.
